# OBJECTCLEAR: COMPLETE OBJECT REMOVAL VIA OBJECT-EFFECT ATTENTION

## ABSTRACT

Object removal requires eliminating not only the target object but also its associated visual effects, such as shadows and reflections. However, diffusion-based inpainting methods often produce artifacts, hallucinate contents, alter background, and struggle to remove object effects accurately. To overcome these limitations, we present a new dataset for OBject-Effect Removal, named *OBER*, which provides paired images with and without object-effects, along with precise masks for both objects and their effects. The dataset comprises high-quality captured and simulated data, covering diverse objects, effects, and complex multi-object scenes. Building on *OBER*, we propose a novel framework, *ObjectClear*, which incorporates an object-effect attention mechanism to guide the model toward the foreground removal regions by learning attention masks, effectively decoupling foreground removal from background reconstruction. Furthermore, the predicted attention map enables an attention-guided fusion strategy at inference, greatly preserving background details. Extensive experiments demonstrate that *ObjectClear* outperforms existing methods, achieving superior object-effect removal quality and background fidelity, especially in challenging real-world scenarios.

## 1 INTRODUCTION

Recent advances in generative models (Rombach et al., 2022; Podell et al., 2024; Meng et al., 2021), such as GPT-4o, have shown strong capabilities in image editing and are widely adopted in real-world applications. Among these tasks, object removal has become a key task, allowing users to erase unwanted content from images. However, seamlessly removing both the object and its effects (*e.g.*, shadows and reflections) while preserving the background remains a challenging problem.

While diffusion-based methods (Podell et al., 2024; Zhuang et al., 2024; Ju et al., 2024; Li et al., 2025; Sun et al., 2025; Ekin et al., 2024; Jia et al., 2025; Chen et al., 2024) have advanced the object removal task, they still struggle with high-fidelity results. As shown in Figs. 1 and 5, existing methods often leave residual artifacts, hallucinate unwanted content, alter backgrounds details, mistakenly remove other objects, and fail to eliminate objects' effects. This is partly because, in existing models, the input mask typically fully occludes the target object, preventing the model from perceiving its surrounding context and associated effects. In addition, the lack of large-scale and publicly available training datasets for object-effect removal also limits further progress in the field.

Existing object removal datasets can be categorized into *simulated* and *camera-captured* data. (1) *Simulated data.* These datasets are often constructed by copy-pasting objects (Jiang et al., 2025; Li et al., 2024) or using pretrained inpainting models to generate pseudo ground truth (Tudosiu et al., 2024). While this allows low-cost generation of large-scale data, such datasets typically lack object effects such as shadows and reflections, causing models trained on them to struggle with effect removal. (2) *Camera-captured data.* Some works (Sagong et al., 2022; Wei et al., 2025) leverage fixed-viewpoint videos to extract paired frames, but this approach limits the foreground to moving objects only and makes it difficult to ensure background consistency between paired samples. Others (Winter et al., 2024; Yu et al., 2025; Yang et al., 2025) capture image pairs before and after object removal, but these datasets are often not publicly available and small in scale due to the high costs.

Achieving *complete removal of objects and effects* while *preserving background consistency* remains challenging, especially with limited high-quality training data. In this study, we focus on both dataset construction and network design (training and inference), explicitly targeting these two goals.

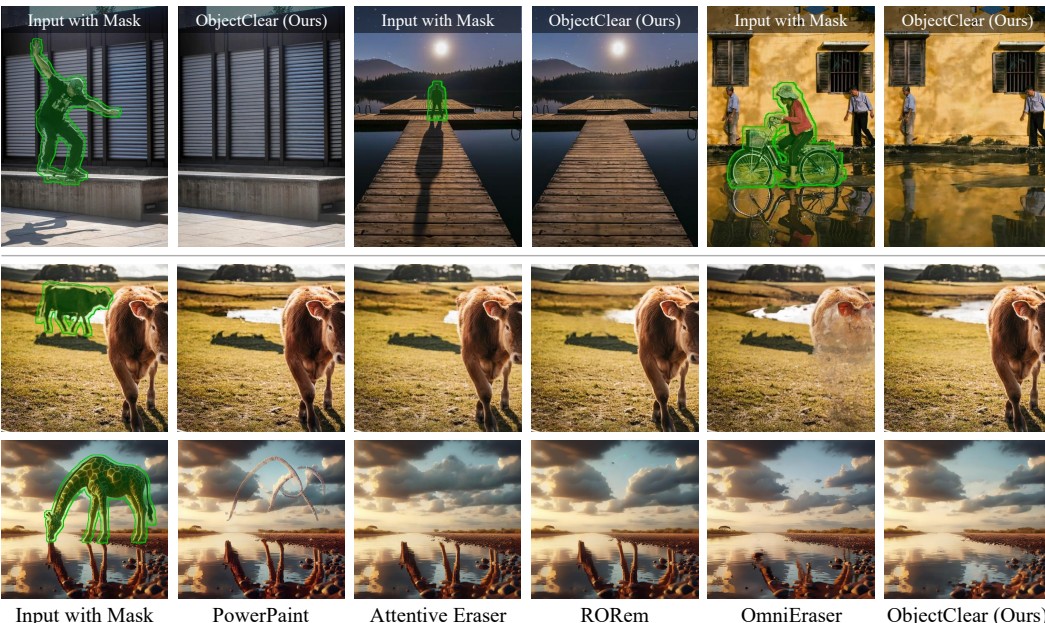

Figure 1: **Object Removal Comparison.** Given an object mask, prior methods often leave residual artifacts or hallucinate undesirable content, change background, and typically fail to remove associated effects such as *shadows* and *reflections*. In contrast, our *ObjectClear* precisely eliminates both the object and its associated effects, achieving seamless object removal results even in challenging cases.

**Dataset -** we present *OBER*, a hybrid dataset for **OB**ject-**E**ffect **R**emoval, which combines both *camera-captured* and *simulated* data with diverse foreground objects, background scenes, and object effects (*e.g.*, indoor/outdoor shadows and reflections). *For the camera-captured data*, we annotate precise object and object-effect masks, which serve as critical supervision during training. *For the simulated data*, we leverage these masks to compute accurate alpha maps, enabling realistic alpha blending of RGBA object layers with effects and high-quality backgrounds. We further extend the simulation pipeline to include multi-object scenarios, enhancing model robustness in challenging cases involving occlusion and object interactions, as shown in the last two rows of Fig. 5. By combining the realism of captured data with the scalability of simulation, OBER offers a *high-quality*, *large-scale*, and *diverse* dataset with a total of 12,715 training samples. In addition, we introduce two new benchmarks: *OBER-Test* and *OBER-Wild*, to support future research.

**Network -** built upon *OBER*, we propose a novel framework, *ObjectClear. (a) During training*, we introduces an Object-Effect Attention (OEA) mechanism that adaptively focuses on the foreground removal region (object and its effects), by supervising cross-attention maps with object-effect masks. The OEA module effectively decouples foreground removal from background reconstruction, enabling more precise and complete object elimination. *(b) During inference*, the predicted attention map supports an Attention-Guided Fusion (AGF) module, helping to preserve background details, as shown in Fig. 7. We further propose a Spatially-Varying Denoising Strength (SVDS) strategy to address incomplete object removal and inconsistent background colors caused by a uniform denoising strength (See Fig. 11). These designs enable the model to adaptively handle removal regions, achieving precise and complete object-effect removal while preserving background details.

Our contributions are summarized as follows:

- We propose *OBER*, a high-quality and large-scale hybrid dataset for object removal, featuring diverse objects, fine-grained annotations of object-effect masks, and complex multi-object scenarios across both simulated and camera-captured settings.
- We introduce a novel framework, *ObjectClear*, which incorporates an Object-Effect Attention (OEA) mechanism that adaptively focuses on foreground removal regions, together with an Attention-Guided Fusion (AGF) and a Spatially-Varying Denoising Strength (SVDS) strategy, thereby improving object removal quality and background fidelity.
- Our approach achieves superior performance on all benchmarks, outperforming existing approaches in terms of both quantitative metrics and visual quality.

## 2 RELATED WORK

**Image Inpainting.** Image inpainting is a long-standing visual editing task that aims to seamlessly reconstruct pixels within a given mask. Early approaches predominantly adopt generative adversarial networks (GANs) (Goodfellow et al., 2014), but often suffer from limited realism and diversity (Liu et al., 2020; Pathak et al., 2016; Ren et al., 2019; Zeng et al., 2019). With the rapid advances in diffusion models (Rombach et al., 2022; Song et al., 2020; Ho et al., 2020; Podell et al., 2024), many methods (Avrahami et al., 2022; Lugmayr et al., 2022; Meng et al., 2021; Zhuang et al., 2024; Ju et al., 2024; Yang et al., 2023) have begun leveraging their strong generative priors to synthesize high-fidelity content, achieving state-of-the-art results in image inpainting. In this work, we adapt the SDXL-Inpainting model (Podell et al., 2024) for photorealistic completion. However, despite their strong generative capabilities, existing inpainting models often lack awareness of object-induced effects (*e.g.*, shadows and reflections), leading to incomplete or inconsistent object removal results.

**Object Removal.** Object removal is a specialized branch of image inpainting that requires explicit consideration of object effects to achieve complete removal, a task currently dominated by diffusion-based models (Winter et al., 2024; Jiang et al., 2025; Liu et al., 2025; Yu et al., 2025; Li et al., 2025; Chen et al., 2025; Zhuang et al., 2024; Ekin et al., 2024; Chen et al., 2024; Yang et al., 2025; Jia et al., 2025; Sun et al., 2025; Wei et al., 2025). A common strategy involves curating high-quality triplet datasets. For instance, ObjectDrop (Winter et al., 2024) builds a real-world dataset by having photographers capture the same scene before and after the removal of a single object. However, its limited scale and lack of public availability hinder broader adoption. To address data scalability, methods like SmartEraser (Jiang et al., 2025) and Erase Diffusion (Liu et al., 2025) rely on synthetic datasets using segmentation or matting techniques to extract foreground objects, but these typically lack annotations of object effects, limiting the models' ability to remove shadows or reflections. To enhance realism, methods such as LayerDecomp (Yang et al., 2025) and OmniPaint (Yu et al., 2025) create costly camera-captured data. OmniPaint (Yu et al., 2025) takes one step further to annotate unlabeled images by a model trained on small-scale real data, while RORem (Li et al., 2025) involves human annotators to ensure the quality of annotated data. In parallel, works like RORD (Sagong et al., 2022) and OmniEraser (Wei et al., 2025) scale data generation by mining realistic video frames with fixed viewpoints, selectively pairing frames with and without target objects while preserving natural object effects. To eliminate reliance on curated datasets altogether, other methods opt for test-time optimization (Chen et al., 2024; Sun et al., 2025; Jia et al., 2025) or providing plug-and-play solutions (Ekin et al., 2024). However, these models rely on implicitly learning object effects without explicitly modeling effect maps, which makes it difficult to maintain background consistency.

## 3 METHODOLOGY

To achieve accurate removal of target objects along with their associated visual effects, such as shadow and reflection, we propose a comprehensive framework consisting of a data curation pipeline (OBER Dataset, Sec. 3.1) and a dedicated object removal model (ObjectClear, Sec. 3.2).

### 3.1 OBER DATASET

ObjectDrop (Winter et al., 2024) shows that even small-scale datasets, when captured with careful camera control over scene consistency, can significantly enhance model generalizability in object removal tasks. However, due to the lack of public access to such datasets and the high cost of real-world data collection, subsequent works often face limitations in dataset scale. To address this challenge, we introduce the *OBER* dataset (**OB**ject-**E**ffect **R**emoval), designed to balance data realism and scalability. As illustrated in Fig. 2, OBER is a *hybrid* dataset consisting of two parts: (1) a small set of camera-captured images adhering to physical realism, and (2) a larger set of simulated images generated by compositing foreground objects, extracted from the real data in (1), onto diverse background scenes.

**Camera-Captured Data.** *(1) Capture Paired Images.* Following the approach of ObjectDrop (Winter et al., 2024), we use fixed cameras to construct a counterfactual dataset by capturing each scene before and after the removal of a single object, while keeping all the other factors unchanged. For each pair, the image with the object is used as the input $I_{in}$, and the image without the object serves as the ground truth $I_{gt}$. In total, we collected 2,878 such counterfactual pairs with 2,715 for training

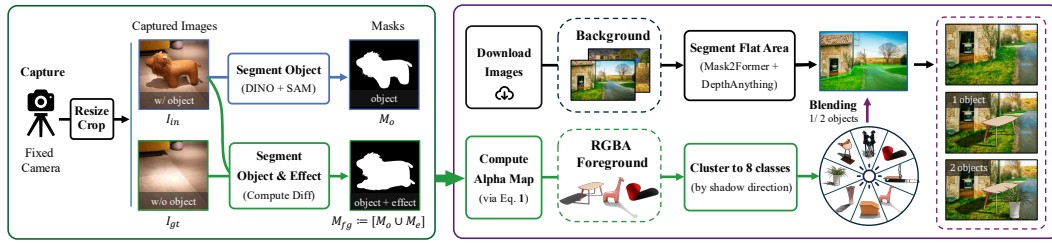

(a) Camera-Captured Data  (b) Simulated Data

Figure 2: **Dataset Construction Pipeline of *OBER*.** The dataset combines both *camera-captured* and *simulated* data, featuring diverse foreground objects and background scenes. It provides rich annotations, including object masks, object-effect masks, transparent RGBA object layers, and complex multi-object scenarios for training and evaluation. (See sample data in Fig. 8)

and 163 for testing. Our dataset encompasses a wide variety of everyday objects commonly found in indoor (e.g., benches, cups, dolls) and outdoor scenes (e.g., pedestrians, vehicles). These image pairs also preserve visual effects such as shadows and reflections. To reduce potential pixel-level misalignment, we downsample and crop all images to a fixed training resolution (i.e., $512 \times 512$). *(2) Segment Object Mask $M_o$.* To obtain the object mask $M_o$ as network input alongside the input image $I_{in}$, we apply off-the-shelf detection and segmentation models, such as DINO (Zhang et al., 2023) and SAM (Kirillov et al., 2023) to $I_{in}$. *(3) Segment Object-Effect Mask $M_o \cup M_e$.* Different from previous methods, we propose to explicitly model the object effects. Therefore, we introduce an object-effect mask $M_{fg}$ that covers both the object $M_o$ and its associated visual effects $M_e$ in our dataset. Unlike the coarse object-effect masks provided in the RORD dataset (Sagong et al., 2022), we construct $M_{fg}$ in an efficient and accurate way by computing the pixel-wise difference between $I_{in}$ and $I_{gt}$. Pixels with differences above a predefined threshold are regarded as part of the object-effect mask. This object-effect mask provides crucial supervision during training, allowing the network to adaptively learn to focus on the target removal regions, including both the object itself and its associated effects.

**Simulated Data.** With the high-quality real data collected, we further scale up the dataset with a carefully designed simulation pipeline. *(1) Collect Background Images.* We begin by downloading high-quality background images from the Internet, which offer a diverse range of background scenes. To select backgrounds with flat regions suitable for object placement, we first apply Mask2Former (Cheng et al., 2022) to segment flat areas corresponding to semantic classes such as "*road, sidewalk, grass, floor*", as illustrated in Fig. 2(b). We then refine the selection by computing the gradient of the depth map generated by Depth Anything V2 (Yang et al., 2024), filtering out regions with significant depth variation to ensure that the inserted objects are placed on visually flat surfaces. *(2) Collect Foreground Objects with Effects.* Based on the camera-captured paired data $(I_{in}, I_{gt})$, along with the object mask $M_o$ and effect mask $M_e$, we compute the alpha map of the foreground object with effects $I_{oe}$ using Eq. 1, where $\varepsilon$ is a small constant added to avoid division by zero. For subsequent compositing, we manually categorize the foreground objects into eight groups based on their shadow direction (Fig. 2(b)).

$$
\alpha(p) = \begin{cases} 0, & \text{if pixel } p \in \text{background area } \overline{M_o} \cap \overline{M_e} \\ 1, & \text{if pixel } p \in \text{object area } M_o \\ (I_{gt} - I_{in})/(I_{gt} + \varepsilon), & \text{if pixel } p \in \text{effect area } M_e \end{cases} \tag{1}
$$

*(3) Blend Objects with Backgrounds.* We randomly sample a background image $I_{bg}$ and a foreground object image with effects $I_{oe}$, and synthesize a composite image using alpha blending: $I_{comp} = (1-\alpha) \cdot I_{bg} + \alpha \cdot I_{oe}$. Beyond single-object cases, we also synthesize *multi-object* data by compositing multiple foreground objects with the same lighting directions, thereby covering scenarios involving object occlusions, as illustrated in Fig. 2(b). These designs ensure physically plausible placement and consistent lighting, reducing the domain gap between simulated and real data. Synthetic data also overcomes challenges in real-world collection, such as limited scale and complex cases (e.g., multi-object occlusions, reflections), while its controllability expands coverage and diversity. In total, we generate 10,000 composite images, significantly enriching the diversity and scalability of our dataset. In particular, the simulation of multi-object compositions leads to notable improvements in object removal robustness and background preservation, as discussed and compared in Table 2.

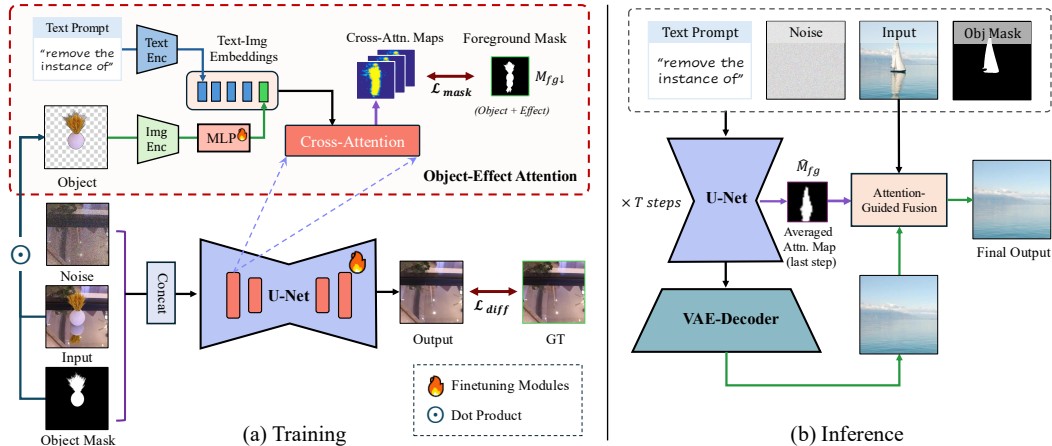

Figure 3: **The Framework of *ObjectClear*.** Given an input image and a target object mask, *Object-Clear* employs an Object-Effect Attention mechanism to guide the model toward foreground removal regions by learning attention masks. The predicted mask further enables an Attention-Guided Fusion strategy during inference, which substantially preserves background details.

## 3.2 OBJECTCLEAR

Our ObjectClear is built upon SDXL-Inpainting (Podell et al., 2024). While SDXL-Inpainting uses a noise map $z_t$, a masked image $I_m$, a corresponding mask $M_o$, and a text prompt $c$ as inputs, we feed the original image $I_{in}$ instead of $I_m$ into the model, and our inputs are expressed as a tuple $< z_t, I_{in}, M_o, c >$. This design encourages the model to better attend to the effect of the target object by leveraging its visual features. Moreover, it facilitates more effective utilization of background information behind the object when transparent objects are to be removed, such as glass cups. To achieve precise and complete object removal while enhancing background preservation, we introduce Object-Effect Attention (OEA), Attention-Guided Fusion (AGF), and Spatially-Varying Denoising Strength (SVDS) strategy, which explicitly attend to object-effect regions.

**Object-Effect Attention.** To enable the model to better attend to both the target object region and its associated effect regions, We integrate text and object image as an object prompt for the cross-attention layers in the base model. Specifically, the text prompt is expressed as "*remove the instance of*", and the visual object is obtained by applying a dot product between the input image $I_{in}$ and $M_o$, as illustrated in Fig. 3(a). These two modalities are then encoded into text embeddings and visual embeddings using the CLIP (Radford et al., 2021) text and vision encoders, respectively. To unify their representation spaces, the visual embeddings are further projected into the same dimensional space as the text embeddings using a lightweight trainable MLP composed of two linear layers. The resulting text embeddings and projected object embeddings are then stacked and used as guidance in the cross-attention blocks of the base model. To encourage the model to focus more accurately on the object and its effect regions, we introduce a mask loss $\mathcal{L}_{mask}$. Concretely, we extract the cross-attention maps corresponding to the visual embedding tokens and denote them as $\mathbf{A}$, which we supervise with the annotated foreground object-effect masks $M_{fg}$ from our OBER dataset. $\mathcal{L}_{mask}$ is designed to minimize the attention values in the background regions while maximizing those in the foreground. This objective can be formulated as:

$$\mathcal{L}_{mask} = \text{mean}(\mathbf{A}[1 - M_{fg}]) - \text{mean}(\mathbf{A}[M_{fg}]), \qquad (2)$$

where $[\cdot]$ denotes indexing operator, and $\mathbf{A}[M_{fg}]$ refers to the attention values within the foreground regions specified by the mask $M_{fg}$.

**Attention-Guided Fusion.** Interestingly, we observe that applying Object-Effect Attention not only improves ObjectClear's precision in object removal, but also produces attention maps that accurately capture both the object and its effects. Based on this observation, we propose an Attention-Guided Fusion strategy that leverages the predicted attention maps during inference to seamlessly blend the generated result with the original image. Specifically, we extract the first-layer cross-attention map corresponding to the object embedding during inference. This attention map is then upsampled to match the resolution of the original image, forming a soft estimate of the object-effect region. To avoid

edge artifacts during blending, we apply a Gaussian blur to the upsampled attention map, producing a soft-edged object-effect mask. This mask is then used in an alpha-blending operation to fuse the generated image with the original input. This strategy significantly reduces undesired background changes introduced by the diffusion denoising process and VAE reconstruction, achieving high-fidelity object removal. Unlike BrushNet (Ju et al., 2024), which relies on user-provided object masks that usually exclude effect regions, our approach leverages precise object-effect masks adaptively generated by Object-Effect Attention.

**Spatially-Varying Denoising Strength.** In diffusion-based image editing, the initial latent for denoising is typically obtained by adding a certain amount of noise to the input image latent. The denoising strength (DS) DS $\in [0, 1]$ controls the noise level: a larger value injects more noise, thereby pushing the initial latent $z_t$ closer to the pure-noise prior. When DS $= 1.0$, the diffusion process starts entirely from noise and discards information from inputs. As shown in Fig. 11, DS $= 1.0$ achieves complete object removal but may introduce noticeable global color shifts. Conversely, using a slightly lower denoising strength (*e.g.*, DS $= 0.99$) preserves color consistency but may lead to incomplete object removal or hallucinated unwanted objects. Motivated by these observations, we propose Spatially-Varying Denoising Strength (SVDS), which applies DS $= 1.0$ within the masked object region and DS $= 0.99$ in the unmasked background, via re-injecting the background during inference. This strategy achieves complete object-effect removal while maintaining background color consistency and preventing edge artifacts in Attention-Guided Fusion.

## 4 EXPERIMENTS

**Implementation.** Our proposed method is built on SDXL-Inpainting (Podell et al., 2024) and fine-tuned with our OBER dataset with an input resolution of $512 \times 512$. Training is conducted with a total batch size of 32 on 8 A100 GPUs for 100k iterations and a learning rate of 1e-5. All experiment results are attained with a guidance scale of 7.5 and 20 denoising steps.

**Evaluation Data.** We evaluate the object removal performance of our method, ObjectClear, against existing approaches on three test datasets: (1) *RORD-Val*: RORD (Sagong et al., 2022) is a widely used object removal dataset. Each sample contains an image pair (images with and without the target object) and a coarse mask covering both the target object and its associated effect. To avoid duplicated scenarios, we randomly select one image per scene, resulting in a RORD-Val with *343* samples. Since most of the object removal methods, including ObjectClear, require only the object mask as input, we augment RORD-Val with accurate object masks manually to enable a more comprehensive evaluation. (2) *OBER-Test*: As described in Sec. 3.1, we split our collected OBER dataset into a training set and a test set. The test set contains *163* samples, each comprising image pairs along with precise masks for both the object and its effect. (3) *OBER-Wild*: To further assess the performance of different methods on out-of-distribution scenarios in the wild, we collect *302* high-quality images featuring objects with associated effects (shadows or reflections) from the Internet. We annotate the object masks using DINO (Zhang et al., 2023) and SAM (Kirillov et al., 2023), and manually annotate the effect masks. Note that the OBER-Wild set does not include the removal ground truth.

**Evaluation Metrics.** For RORD-Val and OBER-Test, where ground truth is available, we evaluate performance using fidelity metrics: PSNR and PSNR-BG (computed only on background regions), as well as two perceptual metrics: LPIPS (Zhang et al., 2018) and CLIP (Radford et al., 2021) (feature distance). For OBER-Wild, which lacks ground truth, we employ ReMOVE (Chandrasekar et al., 2024) to measure the consistency between the object removal region in the result and the surrounding background of the intput. To better assess visual harmony, we modify original ReMOVE to compare the output's removal region with the input background instead of the output background.

### 4.1 COMPARISONS WITH STATE-OF-THE-ART METHODS

We compare ObjectClear with open-source state-of-the-art methods in both image inpainting and object removal. The inpainting methods include SDXL-INP (Podell et al., 2024), PowerPaint (Zhuang et al., 2024), and BrushNet (Ju et al., 2024). The object removal methods include CLIP-Away (Ekin et al., 2024), DesignEdit (Jia et al., 2025), RORem (Li et al., 2025), FreeCompose (Chen et al., 2024), Attentive Eraser (Sun et al., 2025), and OmniEraser (Wei et al., 2025). Following MULAN (Tudosiu et al., 2024), we use the text prompt "*an empty scene*" for the text-guided baselines when necessary.

**Quantitative Evaluation.** Unlike our ObjectClear, which explicitly handles both the target object and its associated effects, most baseline methods operate only within the masked regions, overlooking

Table 1: **Quantitative Comparisons with State-of-the-Art Methods**. The best and second performances are marked in red and orange, respectively. Although ObjectClear takes only the object mask as input, it outperforms previous methods, even when those methods are provided with masks that cover both the object and its associated effect regions.

| Mask Types | Datasets / Metrics | RORD-Val | | | | OBER-Test | | | | OBER-Wild |
|---|---|---|---|---|---|---|---|---|---|---|
| | | PSNR ↑ | PSNR-BG ↑ | LPIPS ↓ | CLIP ↓ | PSNR ↑ | PSNR-BG ↑ | LPIPS ↓ | CLIP ↓ | ReMOVE↑ |
| Object-Effect Mask | SDXL-INP (Podell et al., 2024) | 19.39 | 22.98 | 0.2432 | 0.1024 | 24.07 | 26.93 | 0.1296 | 0.0579 | 0.6983 |
| | PowerPaint (Zhuang et al., 2024) | 19.87 | 21.98 | 0.2303 | 0.0776 | 26.20 | 27.41 | 0.1243 | 0.0409 | 0.8044 |
| | BrushNet (Ju et al., 2024) | 16.82 | 18.87 | 0.3434 | 0.1692 | 20.96 | 23.69 | 0.2052 | 0.1180 | 0.5358 |
| | DesignEdit (Jia et al., 2025) | 20.69 | 22.49 | 0.2946 | 0.1200 | 26.59 | 27.63 | 0.1777 | 0.0629 | 0.8215 |
| | CLIPAway (Ekin et al., 2024) | 18.87 | 20.78 | 0.3328 | 0.0969 | 25.38 | 26.28 | 0.1039 | 0.0349 | 0.7705 |
| | FreeCompose (Chen et al., 2024) | 19.67 | 21.72 | 0.3316 | 0.1110 | 23.39 | 25.15 | 0.1305 | 0.0629 | 0.7555 |
| | Attentive Eraser (Sun et al., 2025) | 20.33 | 21.98 | 0.2545 | 0.1015 | 27.42 | 29.27 | 0.1114 | 0.0249 | 0.7940 |
| | RORem (Li et al., 2025) | 21.61 | 23.11 | 0.3224 | 0.0767 | 27.23 | 27.95 | 0.1042 | 0.0234 | 0.8164 |
| Object Mask | SDXL-INP (Podell et al., 2024) | 20.23 | 24.83 | 0.2042 | 0.0868 | 22.42 | 25.77 | 0.1428 | 0.0771 | 0.6971 |
| | PowerPaint (Zhuang et al., 2024) | 21.46 | 24.62 | 0.1801 | 0.0648 | 22.76 | 24.67 | 0.1544 | 0.0729 | 0.7699 |
| | BrushNet (Ju et al., 2024) | 18.06 | 23.44 | 0.2757 | 0.1821 | 21.19 | 24.38 | 0.1822 | 0.1123 | 0.6341 |
| | DesignEdit (Ekin et al., 2024) | 22.09 | 24.26 | 0.2501 | 0.1021 | 24.63 | 25.48 | 0.1870 | 0.0788 | 0.8163 |
| | CLIPAway (Jia et al., 2025) | 20.58 | 23.21 | 0.2770 | 0.0785 | 22.32 | 24.05 | 0.1357 | 0.0765 | 0.7705 |
| | FreeCompose (Chen et al., 2024) | 20.39 | 22.91 | 0.3015 | 0.0897 | 22.77 | 24.46 | 0.1393 | 0.0690 | 0.7451 |
| | Attentive Eraser (Sun et al., 2025) | 22.17 | 24.59 | 0.1883 | 0.0643 | 25.70 | 27.08 | 0.1201 | 0.0437 | 0.8080 |
| | RORem (Li et al., 2025) | 22.49 | 24.10 | 0.2943 | 0.0634 | 24.51 | 25.28 | 0.1288 | 0.0460 | 0.8121 |
| | OmniEraser (Wei et al., 2025) | 21.79 | 22.98 | 0.2195 | 0.0542 | 24.44 | 24.87 | 0.1783 | 0.0142 | 0.7655 |
| | **ObjectClear (Ours)** | 26.24 | 29.78 | 0.1157 | 0.0299 | 33.04 | 35.62 | 0.0342 | 0.0103 | 0.8470 |

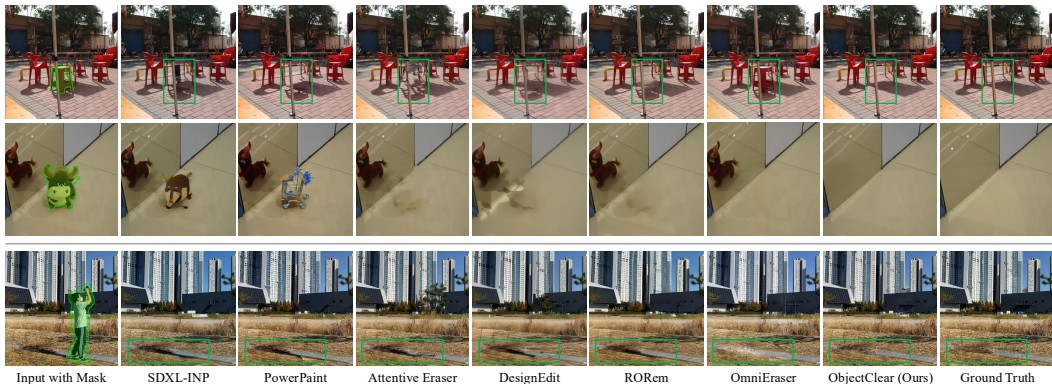

Input with Mask     SDXL-INP     PowerPaint     Attentive Eraser     DesignEdit     RORem     OmniEraser     ObjectClear (Ours)     Ground Truth

Figure 4: **Object Removal on OBER-Test and RORD-Val.** Our ObjectClear effectively removes both the masked objects and their associated effects, including shadows and mirror reflections.

surrounding areas that are visually affected by the object but not explicitly included in the mask. To ensure a fair and comprehensive comparison, we evaluate all methods under two mask conditions: (1) *the input mask covers only the target object*, and (2) *the input mask covers both the target object and its visual effects*. These are referred as Object Mask and Object-Effect Mask settings in Table 1, respectively. As shown in Table 1, ObjectClear achieves state-of-the-art performance across all test sets and metrics. Notably, even when using only the object mask, ObjectClear surpasses methods that rely on both object and effect masks. In particular, it achieves a significant advantage in the PSNR-BG metric, highlighting its superior ability to preserve background consistency with the input.

**Qualitative Evaluation.** Qualitative results are shown in Fig. 4 and Fig. 5. Generation-based inpainting approaches, such as SDXL-INP (Podell et al., 2024), and PowerPaint (Zhuang et al., 2024), often generate new objects within the masked regions while fail to remove the effects of the removed objects. In contrast, previous object removal methods, such as Attentive Eraser (Sun et al., 2025), DesignEdit (Jia et al., 2025), and RORem (Li et al., 2025) demonstrate strong performance in removing the target object itself, but still fail to eliminate the associated effects. OmniEraser shows the capability to remove both objects and their effects, but certain limitations remain: it may not completely remove effects, occasionally erase non-target objects, or alter background details.

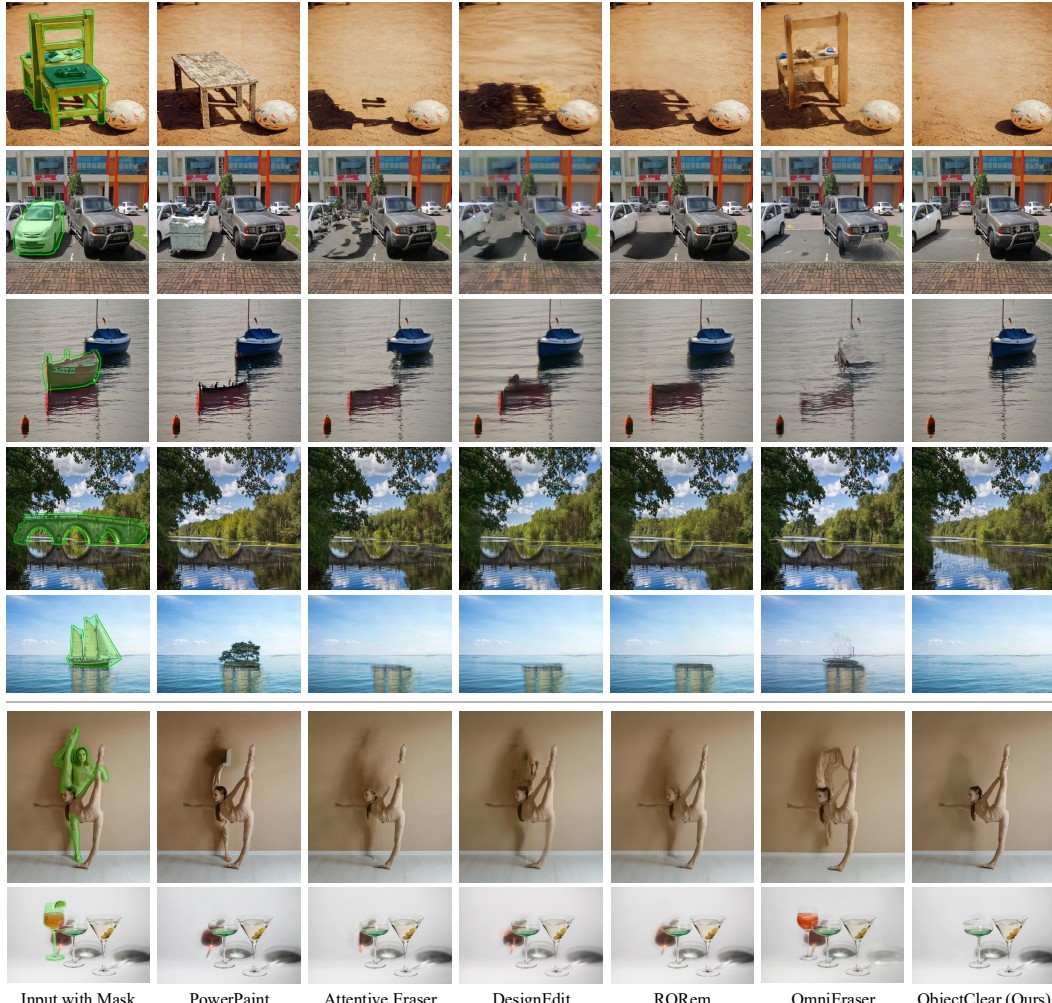

| Input with Mask | PowerPaint | Attentive Eraser | DesignEdit | RORem | OmniEraser | ObjectClear (Ours) |
|---|---|---|---|---|---|---|

Figure 5: **Object Removal on OBER-Wild.** Our ObjectClear not only effectively removes single objects with their shadows and reflections (top five samples), but also accurately removes the target object when multiple mutually occluding objects exist (bottom two samples).

Thanks to our carefully curated training data and the proposed object-effect attention mechanism, ObjectClear effectively removes not only the target object but also its associated shadows and reflections. Moreover, as shown in the last two rows of Fig. 5, ObjectClear effectively handles complex cases involving multiple mutually occluding objects.

## 4.2 ABLATION STUDIES

**The Effectiveness of the Object-Effect Mask Loss $\mathcal{L}_{mask}$.** The object-effect mask loss is designed to explicitly guide ObjectClear to attend to both the target object and its effects, leading to more complete and precise removal. From Table 2 (a) to (b), improvements can be seen across all metrics with $\mathcal{L}_{mask}$ added, indicating its effectiveness in improving object removal accuracy (see visual comparison in Fig. 9). Besides, as shown in the plot of Fig. 6 (left), the predicted object-effect attention map achieves a high recall of 0.97. In Fig. 6 (right), the attention map from the final denoising step (Step 20) effectively reveals object's shadow (red box) and mirror reflection (green box), altogether leading to complete and visually coherent object removal results.

**The Effectiveness of Simulated Data.** The simulated data effectively enhances the scale and diversity of the training data, leading to notable improvements in object removal performance and better background preserving, as shown in Table 2(c). The visual comparisons are provided in Fig. 10. Our carefully designed simulation pipeline, especially the multi-object data synthesis, further enables ObjectClear to tackle challenging scenarios involving multiple occluded objects with intersecting effects, as illustrated in the last two rows of Fig. 5.

Table 2: **Quantitative Results of Ablation Studies.** Based on our camera-captured data (CC Data), object removal performance improves progressively by adding the object-effect mask loss ($\mathcal{L}_{mask}$) and simulated training data (Sim. Data). Further improvement is achieved through the proposed Attention-Guided Fusion (AG Fusion) and Spatially-Varying Denoising Strength (SVDS) strategy.

| Exp. | CC Data | $\mathcal{L}_{mask}$ | Sim. Data | AG Fusion | DS | PSNR ↑ | PSNR-BG ↑ | LPIPS ↓ | CLIP ↓ |
|------|---------|---------------------|-----------|-----------|-----|--------|-----------|---------|--------|
| (a) | ✓ | | | | 0.99 | 27.29 | 27.96 | 0.0910 | 0.0247 |
| (b) | ✓ | ✓ | | | 0.99 | 27.56 | 28.37 | 0.0845 | 0.0217 |
| (c) | ✓ | ✓ | ✓ | | 0.99 | 28.04 | 28.80 | 0.0805 | 0.0196 |
| (d) | ✓ | ✓ | ✓ | ✓ | 0.99 | 32.77 | 35.50 | 0.0348 | 0.0106 |
| (e) | ✓ | ✓ | ✓ | ✓ | 1.00 | 31.49 | 33.46 | 0.0375 | 0.0120 |
| (f) | ✓ | ✓ | ✓ | ✓ | SVDS | **33.04** | **35.62** | **0.0342** | **0.0103** |

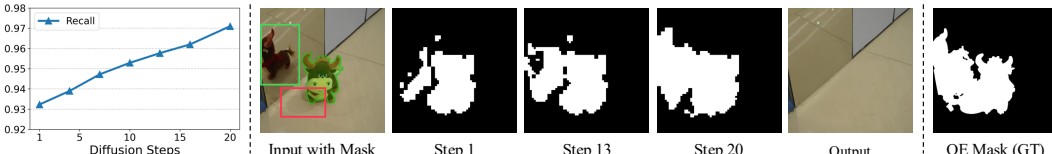

Figure 6: **Object-Effect Attention Map.** ObjectClear achieves a relatively high recall in the object-effect attention map, with recall values increasing as the denoising step progresses. The attention map obtained in the final step effectively covers both the target object and its associated effects, including the object's shadow (red box) and its reflection in the mirror (green box).

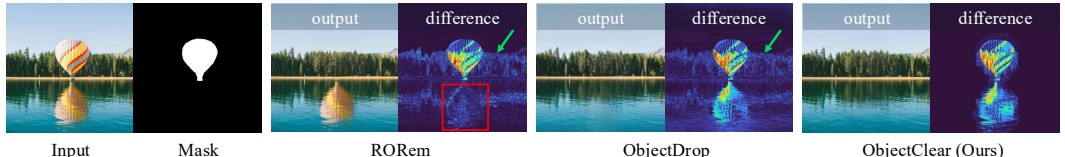

Figure 7: **Effectiveness of Attention-Guided Fusion.** The difference maps visualize the pixel-wise differences between input and output. RORem (Li et al., 2025) fails to remove the reflection and alters the background, yielding low differences in the reflection area and high values in the background. In contrast, both ObjectDrop (Winter et al., 2024) and our ObjectClear successfully remove the reflection, with ObjectClear better preserving the background as indicated by lower differences.

**Effectiveness of the Attention-Guided Fusion.** The attention-guided fusion is designed to preserve background consistency after object removal, a common challenge in generation-based methods like ObjectDrop (Winter et al., 2024) and RORem (Li et al., 2025), as illustrated in Fig. 7. As the denoising process, our object-effect attention yields increasingly accurate attention maps (Fig.6), which we leverage to guide the fusion of input background. This strategy significantly improves background preservation, as evidenced by large gains in PSNR-BG. Consequently, we observe a marked boost in overall performance, with PSNR increasing from 28.04 to 32.77, as shown in Table 2(c–d).

**Effectiveness of the Spatial-Varying Denoising Strength.** The proposed Spatially-Varying Denoising Strength (SVDS) addresses two major issues: incomplete object removal or hallucinated objects when using DS = 0.99, and global color shifts when using DS = 1.0, as illustrated in Fig. 11. As shown in Table 2(d–f, SVDS (f) outperforms both DS = 0.99 (d) and DS = 1.0 (e). It effectively facilitates complete object removal while maintaining consistent background colors and details.

## 5 CONCLUSION

We introduce *ObjectClear*, a practical framework for object removal that achieves high-quality object-effect removal while maintaining background consistency across diverse real-world scenarios. The framework employs an Object-Effect Attention to adaptively focus on removal regions, along with Attention-Guided Fusion and Spatial-Varying Denoising Strength strategy to preserve background details. In addition, we present *OBER*, a large-scale and diverse hybrid dataset that integrates camera-captured and simulated data. Thanks to our dataset and network designs, ObjectClear demonstrates robust and superior performance, effectively overcoming key challenges in object removal task.

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

APPENDIX

In this supplementary material, we provide additional discussions and results to complete the main paper. In Sec. A, we present further details on data augmentation during training and other extended applications. In Sec. B, we provide more details of the proposed OBER dataset, including statistics and some examples for demonstration. In Sec. C, we report additional results, such as more results on ablation studies, a user study, experiments with user strokes, more results of object insertion and movement, and additional comparisons on in-the-wild data.

## A  MORE DETAILS OF OBJECTCLEAR

### A.1  TRAINING AUGMENTATIONS

During training, we apply on-the-fly random cropping to enable the network to learn object removal across varying object sizes. We also apply color augmentation and random flipping to enhance model robustness. To improve the model's robustness to the estimated or user-provided coarse mask, several previous methods (Winter et al., 2024; Jiang et al., 2025; Wei et al., 2025) have introduced mask augmentation techniques. In line with these approaches, we also apply dilation and erosion to the input mask during training. Different from previous practice, our method employs an *object-aware* dilation and erosion strategy, where the dilation and erosion kernel size is adaptively determined based on the size of the object. As demonstrated in the qualitative results in Sec. C.6, our method effectively handles coarse user-drawn masks by implicitly completing and refining them, showcasing strong robustness to imprecise inputs.

### A.2  MORE DETAILS FOR EXTENDED APPLICATIONS

ObjectClear can be flexibly extended to various applications. In this section, we provide the implementation details of object insertion and object movement.

**Object Insertion.** The insertion network also leverages the OBER dataset for training and adopts the same architecture as ObjectClear (Fig. 3 of the main paper), thus receiving the input tuple of $< z_t, I_{in}, M_o, c >$ and supervised by $I_{GT}$ (for output image) and $M_{fg}$ (for object-effect attention). However, $I_{in}$, $I_{GT}$, and $c$ are constructed in an "reverse" way compared with the removal network.

While the ground truth $I_{GT}$ is the original image with the object accompanied by natural effects, $I_{in}$ is the one with the object simply copied and pasted on the background. Specifically, to generate $I_{in}$, we first extract the object from $I_{GT}$ with the object mask $M_o$ only, and then we paste it onto the corresponding background image. In addition to the image pair, the input text $c$ is also reversed to "*insert the instance of*". Notably, the object-effect attention map is still supervised by $M_{fg}$. In this context, $M_{fg}$ refers to the mask that covers both the object and its *generated* effects (e.g., shadows or reflections).

During inference, we directly paste an object (without effects) onto the background scene, along with its object mask $M_o$, which forms the input image $I_{in}$. The network then generates the output image where both the object and its generated effects are harmoniously inserted. Since the insertion network also integrates the object-effect attention, we can also apply Attention-Guided Fusion to preserve the background fidelity while generating realistic object effects.

**Object Movement.** To enable object movement, we combine the ObjectClear and insertion network introduced above. Specifically, we first apply ObjectClear to remove the target object along with its associated effects, resulting in a clean object-free background. The object is then extracted using its given object mask. Users are allowed to specify a new location and optionally adjust the object scale before the object is re-inserted into the clean background. With our insertion network, the object is harmonized with the new context by generating realistic effects. This two-stage approach supports controllable object movement while ensuring visual realism and consistency.

We provide the visual results of object insertion and object removal in Fig. 16, ObjectClear is capable of generating natural visual effects accordingly.

Table 3: **Dataset Overview.** The OBER dataset consists of camera-captured and simulated training data, as well as two testing sets: OBER-Test (w/ ground truth) and OBER-Wild (w/o ground truth). All data subsets are annotated with object masks and object-effect masks.

| Properties | Training Set | | Testing Set | |
|---|---|---|---|---|
| | Camera-Captured | Simulated | OBER-Test | OBER-Wild |
| #Image Pairs | 2,715 | 10,000 | 163 | 302 |
| w/ Ground Truth | ✓ | ✓ | ✓ | ✕ |
| w/ Object Mask | ✓ | ✓ | ✓ | ✓ |
| w/ Object-Effect Mask | ✓ | ✓ | ✓ | ✓ |

Table 4: **Comparison of OBER Dataset with Existing Datasets.** ∗ indicates that the dataset is not publicly available.

| | Description | RORD (Sagong et al., 2022) | MULAN (Tudosiu et al., 2024) | DESOBAv2 (Liu et al., 2024) | Counterfactual∗ (Winter et al., 2024) | Video4Removal∗ (Wei et al., 2025) | OBER (ours) |
|---|---|---|---|---|---|---|---|
| Tasks | Object Removal | ✕ | ✓ | ✕ | ✓ | ✓ | ✓ |
| | Effect Removal | ✕ | ✕ | ✓ | ✓ | ✓ | ✓ |
| | Object-Effect Removal | ✓ | ✕ | ✕ | ✓ | ✓ | ✓ |
| Annotations | Object Mask | ✕ | ✓ | ✓ | ✓ | ✓ | ✓ |
| | Effect Mask | ✕ | ✕ | ✓ | ✕ | ✕ | ✓ |
| | Object-Effect Mask | ✓ (Coarse) | ✕ | ✓ | ✕ | ✕ | ✓ |
| | RGBA Objects | ✕ | ✕ | ✕ | ✕ | ✕ | ✓ |
| | Multi Objects | ✓ | ✓ | ✕ | ✓ | ✓ | ✓ |
| | Camera-Captured GT | ✓ | ✕ | ✕ | ✓ | ✓ | ✓ |

# B  MORE DETAILS OF OBER DATASET

**Overview.** Table 3 provides an overview of our OBER dataset. The training set consists of 2,715 camera-captured image pairs and 10,000 simulated pairs, all annotated with object masks and object-effect masks. For evaluation, we provide two test subsets: OBER-Test (includes 163 pairs with ground truth) and OBER-Wild (includes 302 pairs without ground truth), where object masks and object-effect masks are also available. In addition, we showcase some samples from our OBER dataset in Fig. 8, including camera-captured data and simulated data, alongside with their annotations such as object masks, object-effect masks, and RGBA object foregrounds.

**Reflection Pair Simulation.** Thanks to the strong priors of generative models, we observed that the model trained without reflection data was still able to remove some simple reflection effects, a finding consistent with the conclusions drawn in ObjectDrop (Winter et al., 2024). However, the model often struggled with more challenging reflection removal cases, such as when the object and its reflection were spatially separated or when the reflection was heavily distorted by surface ripples. To address this limitation, we adopted a human-in-the-loop strategy to collect paired reflection data. Specifically, we used our trained model to perform inference on 200 real-world reflection images and manually selected 50 high-quality results (as shown in Fig. 8(b)), which were then added as an important supplement to the training data. We found that even a small amount of high-quality reflection data pairs could considerably improve the model's ability to generalize across diverse reflection scenarios.

**Comparison with Existing Datasets.** We compare our OBER dataset with existing datasets, including those focused on object removal (RORD (Sagong et al., 2022), MULAN (Tudosiu et al., 2024), Counterfactual Dataset (Winter et al., 2024)) and one on shadow removal (DESOBA-v2 (Liu et al., 2024)). As summarized in Table 4, our OBER is the only dataset that includes all three types of mask annotations and RGBA foreground objects, enabling a wide range of tasks such as object removal, effect removal, and joint object-effect removal.

While RORD (Sagong et al., 2022) is among the first to provide object-effect masks, those masks are *coarse* without separate annotations for objects and their effects. As a result, it cannot support independent removal of either the object or the effect, but only their joint removal. In contrast, our OBER dataset provides precise and separate masks for objects and their effects, enabling more flexible and fine-grained removal tasks. In addition, MULAN (Tudosiu et al., 2024) focuses solely on object

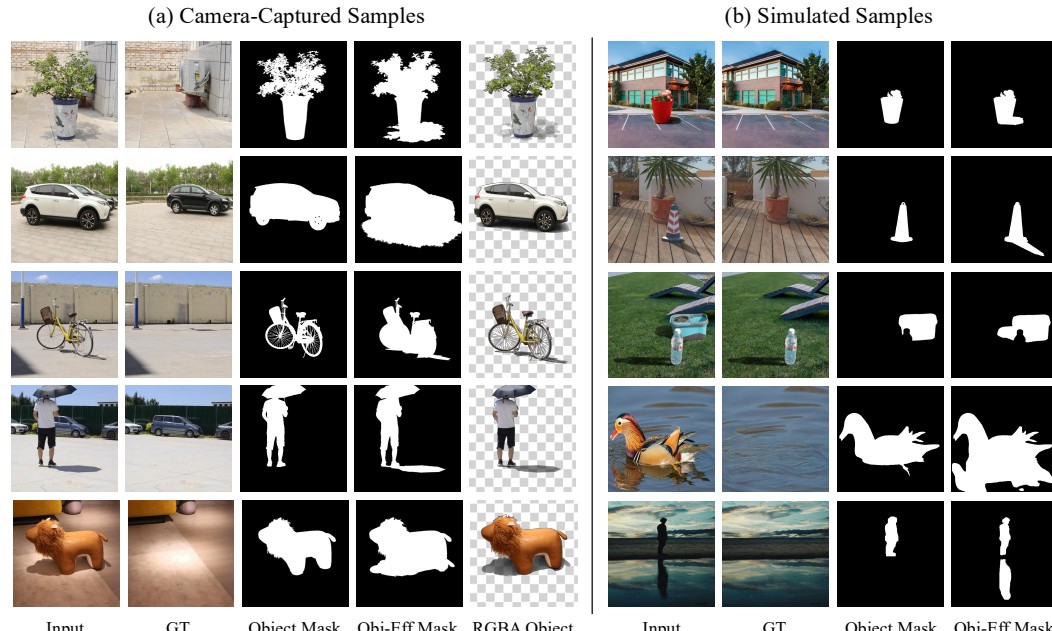

|  | (a) Camera-Captured Samples |  |  |  |  | (b) Simulated Samples |  |  |
|---|---|---|---|---|---|---|---|---|
| Input | GT | Object Mask | Obj-Eff Mask | RGBA Object |  | Input | GT | Object Mask | Obj-Eff Mask |

Figure 8: **Samples from our OBER Dataset.** We provide some samples from our OBER dataset including both camera-captured data and simulated data. While all data from both categories are annotated with fine-grained object masks and object-effect masks, we also extract RGBA foreground objects with associated effects from camera-captured data (as discussed in Sec. 3.1 in the main paper), which could be used to construct realistic simulated samples. In the simulated samples, we not only include the shadow effects but also the reflection effects, which are rarely included in other datasets (Table 4), thanks to our reflection pair simulation strategy detailed in Sec. B.

removal without addressing associated effects such as shadows or reflections. DESOBA-v2 (Liu et al., 2024) targets only effect removal and thus does not handle the objects. Moreover, the ground truth images in both datasets are not captured by cameras but are instead synthesized using generative models, which may limit their realism. The Counterfactual Dataset proposed in ObjectDrop (Winter et al., 2024) is not publicly available, and it only includes object masks without object-effect masks. It also lacks foreground RGBA data for objects and associated effects, limiting its extensibility and applicability to more advanced object editing tasks.

## C  MORE RESULTS

### C.1  RESULTS FOR ABLATION STUDY

**Effectiveness of $\mathcal{L}_{mask}$.** Thanks to the rich annotations in our OBER dataset, the object-effect mask enables a supervision loss, denoted as $\mathcal{L}_{mask}$, which guides the cross-attention layers to focus on both the object and its associated effects, while preserving background textures. This supervision facilitates a decoupled optimization of object removal and background reconstruction. As shown in Fig. 9, with the mask loss $\mathcal{L}_{mask}$, the network can adaptively identify the object-effect regions to be removed, as reflected in the attention mask (shown in yellow boxes). This leads to more accurate and complete removal of objects and their shadow effects, without mistakenly erasing unrelated background content.

**Effectiveness of Simulated Data.** To balance data realism and scalability, in addition to the camera-captured data, we scale up our OBER dataset with a carefully designed simulation pipeline. Our simulation data is generated by compositing the foreground RGBA object (extracted from camera-captured data) onto diverse backgrounds. In particular, the simulation of multi-object compositions leads to notable improvements in object removal robustness when mutually occluding objects exist, as shown in Fig. 10 (*left*). Furthermore, our simulated reflection image pairs (discussed in Sec B) greatly

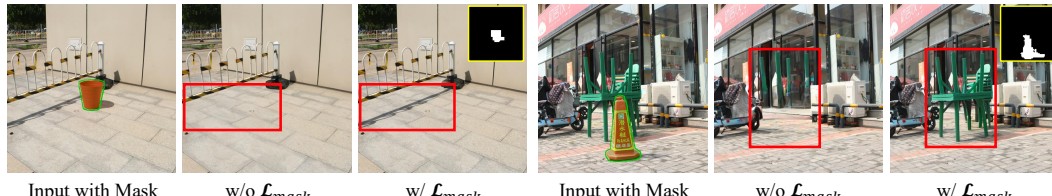

Input with Mask    w/o $\mathcal{L}_{mask}$    w/ $\mathcal{L}_{mask}$     Input with Mask    w/o $\mathcal{L}_{mask}$    w/ $\mathcal{L}_{mask}$

Figure 9: **Effectiveness of $\mathcal{L}_{mask}$.** It could be observed that when training without the $\mathcal{L}_{mask}$ supervision, the model struggles to accurately remove the target object and its effects, leading to mistakenly erasing unrelated background object (*right*) or effects (*left*). In contrast, when supervised with $\mathcal{L}_{mask}$, the attention mask (shown in yellow boxes) could identify the removal regions well, thus leading to more accurate and complete removal results.

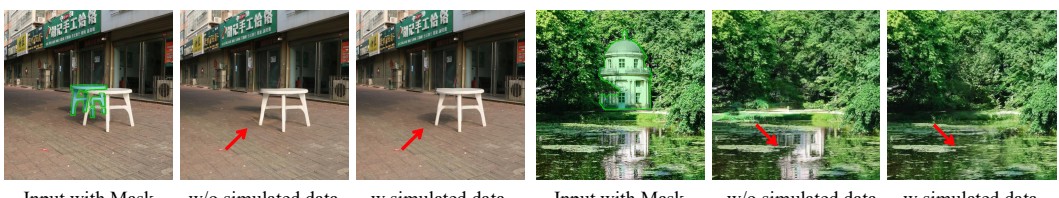

Input with Mask    w/o simulated data    w simulated data     Input with Mask    w/o simulated data    w simulated data

Figure 10: **Effectiveness of Simulated Data.** Since our simulation data includes multi-object compositions, training with such data enables the model to accurately remove the target object and its associated effects while preserving unrelated object effects (*left*). In addition, adding the reflection data pairs during training greatly enhances the model capability of removing reflections, including challenging cases with significant distortion due to water surface ripples (*right*).

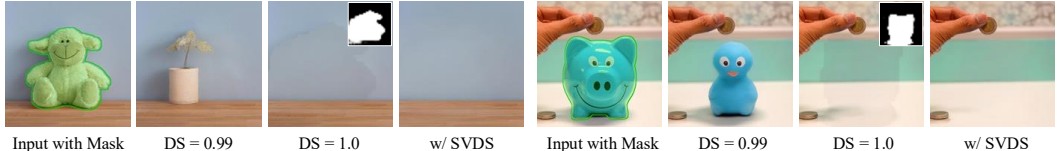

Input with Mask    DS = 0.99    DS = 1.0    w/ SVDS     Input with Mask    DS = 0.99    DS = 1.0    w/ SVDS

Figure 11: **Effectiveness of Spatially-Varying Denoising Strength (SVDS).** DS = 0.99 often leads to incomplete removal or hallucinated objects, while DS = 1.0 causes noticeable color inconsistency (shown after AGF, where the background is from the input image and the object/affected areas are from the removal results). In contrast, SVDS achieves complete object removal with consistent background colors.

improve the model capability of removing reflections, including challenging cases with significant distortion due to water surface ripples, as shown in Fig. 10 (*right*).

**Effectiveness of Attention-Guided Fusion.** The attention map supervised by $\mathcal{L}_{mask}$ supports the Attention-Guided Fusion (AG Fusion) strategy during inference. It helps to blend the generated image with the original input via a copy-and-paste operation, where pixels within the object-effect region are taken from the generated result, and the rest are preserved from the original image. Such practice effectively reduces undesired background detail changes caused by VAE reconstruction errors and the diffusion process, thereby greatly preserving the background fidelity. In Fig. 12, we visualize the background detail changes by showing the difference maps between the generated image and the corresponding input, where a clear improvement on background preservation could be observed.

**Effectiveness of Spatially-Varying Denoising Strength.** As described in the manuscript, in diffusion-based image editing, the initial latent for denoising is typically obtained by adding noise to the input image latent. The denoising strength (DS), DS $\in [0, 1]$, controls the noise level: a larger value injects more noise, thereby pushing the initial noisy latent closer to the pure-noise prior. When DS = 1.0, the diffusion process starts entirely from noise, discarding information from the input. In this paper, we propose Spatial-Varying Denoising Strength (SVDS), which applies DS = 1.0 within the masked object region and DS = 0.99 (an empirical setting commonly adopted by previous methods (Podell et al., 2024)) outside in the unmasked background, ensuring complete object removal while maintaining color consistency. As shown in Fig. 11, setting DS = 0.99 often leads to incomplete

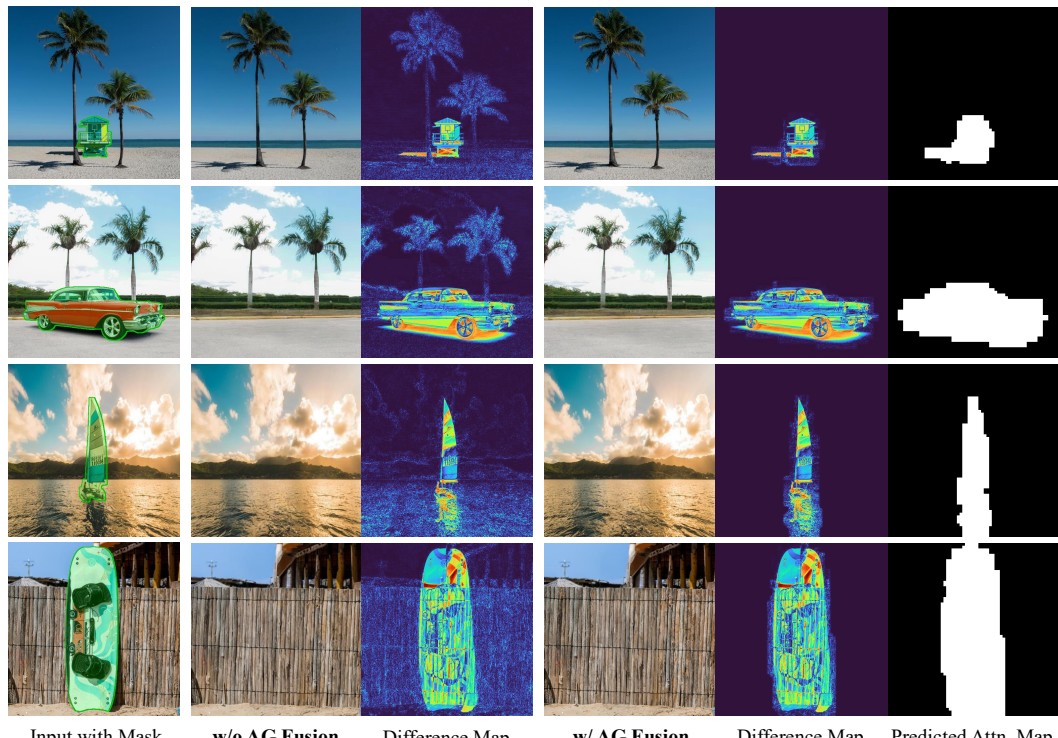

| Input with Mask | **w/o AG Fusion** | Difference Map | **w/ AG Fusion** | Difference Map | Predicted Attn. Map |
|---|---|---|---|---|---|

Figure 12: **Effectiveness of Attention-Guided Fusion (AG Fusion).** We visualize the background detail changes by showing the difference maps between the generated image and the corresponding input. In ideal cases, the difference maps should show large values only in the object removal regions, including the object and its effects, while showing little difference in the unrelated background. It could be observed that when AG Fusion is not employed, although the difference in object removal regions is significant, the difference in the unrelated background is also noticeable. However, with the AG Fusion strategy, we almost eliminate the undesirable background difference thanks to the accurate attention map predicted.

removal or hallucinated objects, whereas DS = 1.0 results in noticeable color inconsistency. To highlight this inconsistency, the results of DS = 1.0 in Fig. 11 are obtained after the Attention-Guided Fusion (AGF) operation, where the background is taken from the original input image, while the object and affected areas are taken from the removal results. In contrast, our method with SVDS achieves superior performance in both object removal and preservation of background color consistency.

## C.2    GENERALIZATION TO MULTI-OBJECT REMOVAL

Although the OBER dataset contains only a small number of multi-object removal cases, we observe that our model generalizes well to multi-object cases (Fig. 13). This can be attributed to the robust generalization capability of the network trained on our dataset.

## C.3    FAIR COMPARISON WITH OBJECT-EFFECT MASK

Our Attention-Guided Fusion (AGF) module leverages object-effect masks predicted by the proposed Object-Effect Attention to blend the original input background back into the generated result. Importantly, these masks are predicted by our model rather than taken from annotations, ensuring that we do not use any privileged information unavailable to other approaches. Furthermore, existing baseline methods do not have the capability to predict object–effect masks and therefore cannot perform background blending in the same way. This makes AGF an integral part of our model design rather than an external post-processing step, and the comparisons in the main paper are therefore fair.

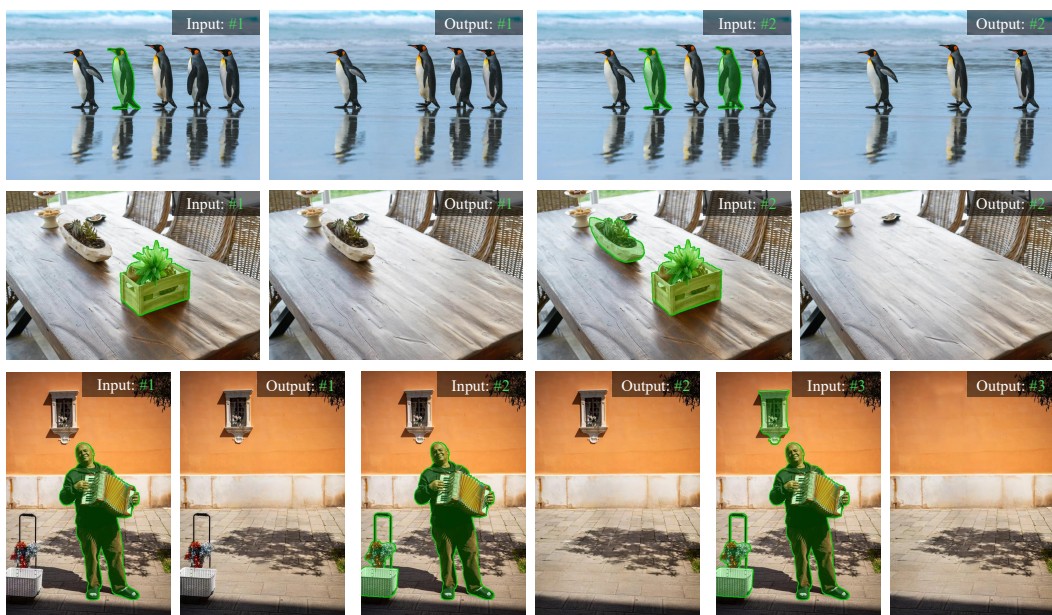

Figure 13: **Multi-object removal.** Our method removes one or multiple objects simultaneously using masks to indicate the targets.

Table 5: **Quantitative results on RORD-Val with object-effect masks for blending.** All baseline methods are equipped with background blending using ground-truth object-effect masks. The best and second performances are marked in red and orange, respectively. Our method using our predicted object-effect masks achieves the best performance across all metrics.

| Method | PSNR↑ | LPIPS↓ | DINO↓ | CLIP↓ |
|---|---|---|---|---|
| SDXL-INP (w/b) | 21.67 | 0.1592 | 0.0688 | 0.0932 |
| PowerPaint (w/b) | 22.51 | 0.1472 | 0.0560 | 0.0606 |
| BrushNet (w/b) | 18.48 | 0.2421 | 0.1572 | 0.1465 |
| DesignEdit (w/b) | 23.73 | 0.1580 | 0.0721 | 0.0755 |
| CLIPAway (w/b) | 21.96 | 0.1735 | 0.0666 | 0.0734 |
| FreeCompose (w/b) | 23.08 | 0.1603 | 0.0829 | 0.0834 |
| Attentive Eraser (w/b) | 22.90 | 0.1700 | 0.0880 | 0.0854 |
| RORem (w/b) | 25.24 | 0.1398 | 0.0387 | 0.0498 |
| **Ours** | 26.24 | 0.1157 | 0.0191 | 0.0299 |

To further demonstrate that the performance gain is not solely due to the background blending, we perform an additional experiment where all baseline methods are given the ground-truth object-effect mask for blending. As shown in Table 5, our method continues to outperform all baselines under this setting, indicating that our superior results primarily come from more effective object–effect generation rather than the availability of blending masks.

C.4    COMPARISONS ON ADDITIONAL BENCHMARKS

To further evaluate the robustness and generalization ability of our method, we conduct two additional benchmarks, *i.e.*, MULAN (Tudosiu et al., 2024) and RemovalBench (Wei et al., 2025), under the object-only mask setting, where only object regions are provided and effect masks are unavailable.

**MULAN Dataset.** The ground truth of some samples in MULAN (Tudosiu et al., 2024) retains shadows, which may introduce bias when evaluating object-effect removal. To enable a fair comparison

Table 6: **Quantitative comparison on Object-Only Mask Benchmark.** The best and second performances are marked in red and orange, respectively.

| Methods | MULAN (500 w/o shadow) | | | | RemovalBench (object mask) | | | |
|---|---|---|---|---|---|---|---|---|
| | PSNR↑ | LPIPS↓ | DINO↓ | CLIP↓ | PSNR↑ | LPIPS↓ | DINO↓ | CLIP↓ |
| SDXL-INP | 19.91 | 0.2494 | 0.1324 | 0.1312 | 21.26 | 0.1968 | 0.0938 | 0.0926 |
| PowerPaint | 21.18 | 0.2449 | 0.1087 | 0.0962 | 22.21 | 0.2001 | 0.0911 | 0.0907 |
| BrushNet | 18.22 | 0.3181 | 0.2062 | 0.1893 | 20.58 | 0.2098 | 0.1182 | 0.1098 |
| DesignEdit | 23.26 | 0.2375 | 0.1114 | 0.0725 | 23.87 | 0.2168 | 0.1020 | 0.0755 |
| CLIPAway | 20.08 | 0.2666 | 0.1180 | 0.1152 | 20.78 | 0.2035 | 0.0934 | 0.0956 |
| FreeCompose | 21.30 | 0.2337 | 0.0828 | 0.0703 | 22.60 | 0.1782 | 0.0733 | 0.0688 |
| Attentive Eraser | 23.96 | 0.1960 | 0.0551 | 0.0397 | 24.77 | 0.1538 | 0.0463 | 0.0388 |
| RORem | 23.53 | 0.2369 | 0.0571 | 0.0438 | 23.70 | 0.1746 | 0.0532 | 0.0416 |
| OmniEraser | 21.56 | 0.2642 | 0.0728 | 0.0682 | 23.83 | 0.1766 | 0.0460 | 0.0481 |
| **Ours** | 24.89 | 0.1586 | 0.0468 | 0.0373 | 27.90 | 0.0942 | 0.0230 | 0.0142 |

focusing solely on object removal, we randomly sampled 500 shadow-free image pairs from MULAN for testing. As shown in Table 6, our method achieves the best performance across all metrics, even surpassing RORem (Li et al., 2025), which was trained on MULAN, thereby demonstrating the strong object removal capability of our approach.

**RemovalBench Dataset.** RemovalBench was proposed by OmniEraser (Wei et al., 2025) as a benchmark for object-effect removal, which aligns with our task setting. For a fair comparison, all methods use their default input sizes, then resize the outputs to the same size (short side 512) for evaluation. Table 6 shows our approach outperforms all baselines across metrics on this dataset.

## C.5  USER STUDY

To enable a more comprehensive evaluation, we conducted a user study on object removal results for in-the-wild images. We compared ObjectClear with three representative state-of-the-art methods: PowerPaint (Zhuang et al., 2024), RORem (Li et al., 2025), and OmniEraser (Wei et al., 2025). For a fair comparison, we evaluated PowerPaint and RORem under two settings: (1) conditioned on the object mask and (2) conditioned on the object-effect mask, while OmniEraser was conditioned only on the object mask. All ObjectClear results were generated using only the object mask, and we compared our outputs against both settings of PowerPaint and RORem, as well as the object-mask setting of OmniEraser.

We invite a total of 30 participants for this user study. Each volunteer was presented 60 randomly selected image quadruples, consisting of: *an input image*, *two results from a baseline method under different mask conditions*, and *our result* (for OmniEraser, only the object-mask result was provided, consistent with our setting, thus forming a triple set). Participants were asked to select the best removal result based on two criteria: the realism of the object region and the preservation of background details. As summarized in Fig 14, ObjectClear outperforms the baselines under both mask settings. Notably, although baseline methods benefited from access to object-effect masks, our ObjectClear won more user preference with the object mask only.

## C.6  RESULTS WITH USER STROKES

In practical applications, users often interact with visual systems through imprecise or casually drawn inputs, such as rough scribbles or incomplete masks. These inputs may vary significantly in shape, location, and accuracy. Therefore, it is essential for a robust object removal network to effectively process such arbitrary mask inputs without relying on carefully crafted annotations. Benefiting from our mask augmentation strategy and object-effect attention mechanism, our network demonstrates strong robustness to diverse mask inputs. In this subsection, we simulate user strokes and feed them into the network along with the images. The resulting outputs and attention maps show that our network can accurately identify and attend to the object and its associated effects, even with imprecise masks, as illustrated in Fig. 15.

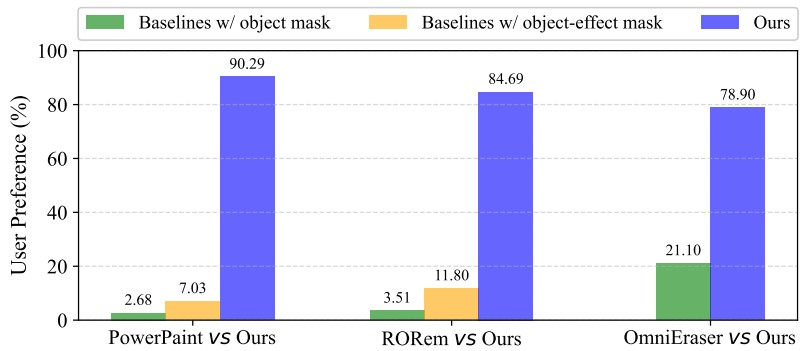

Figure 14: **User Study.** Our ObjectClear is preferred by human voters over three representative state-of-the-art methods (PowerPaint (Zhuang et al., 2024), RORem (Li et al., 2025), and OmniEraser (Wei et al., 2025)). Results of OmniEraser and our method use only the object mask, while PowerPaint and RORem are evaluated with and without the effect mask.

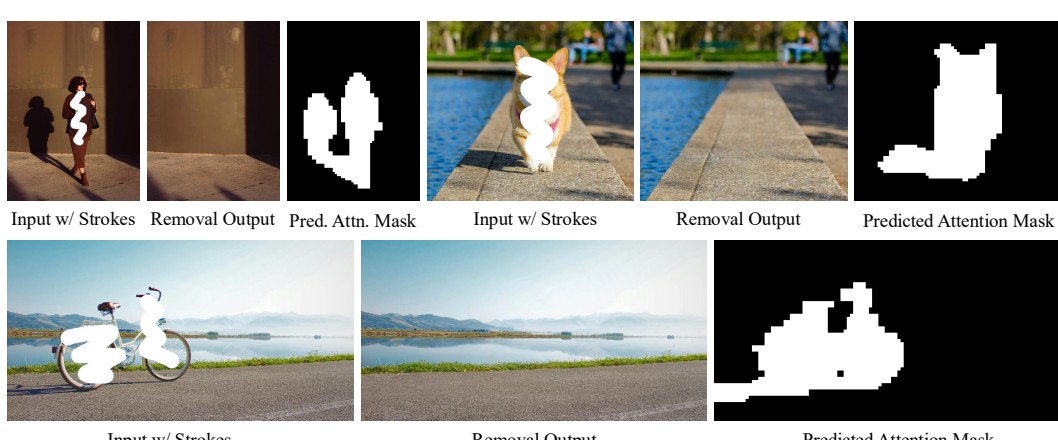

Figure 15: **Results with User Strokes.** We simulate user strokes and feed them into the network along with the images. The resulting outputs and attention maps show that our network can accurately identify and attend to the object and its associated effects, even with imprecise masks.

## C.7 OBJECT INSERTION AND MOVEMENT.

As shown in Fig. 16, even when only the target objects are specified for insertion and movement, ObjectClear is capable of generating plausible and natural shadows and reflections accordingly.

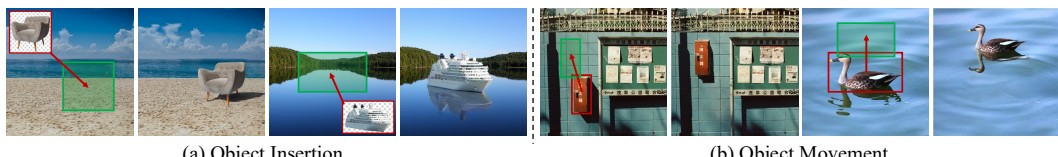

(a) Object Insertion          (b) Object Movement

Figure 16: **Object Insertion and Movement.** In addition to accurately inserting or repositioning objects, our ObjectClear also generates plausible and natural shadows and reflections accordingly.

## C.8 LIMITATIONS

While ObjectClear exhibits strong performance in removing objects and their associated effects, it still faces challenges in highly complex scenarios. Specifically, in cases with overlapping shadows from multiple objects or complex lighting conditions, it can be difficult to disentangle which shadows belong to which objects. As a result, the model may fail to remove the shadows of the target object

(Fig. 17a) or remove shadows of other objects (Fig. 17b). Effectively disentangling object-specific shadows in such complex scenes remains an important direction for future work.

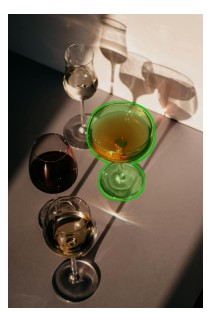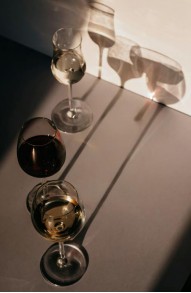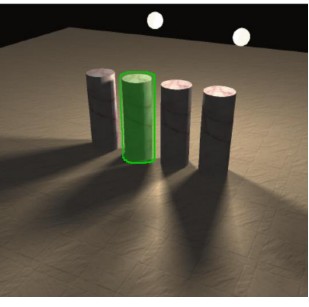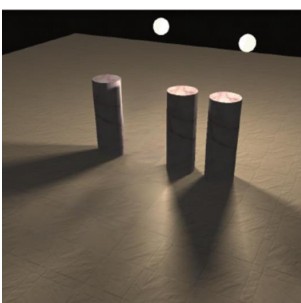

(a) Under-Removal with Effect Interactions        (b) Over-Removal with Effect Interactions

Figure 17: **Limitations.** In complex scenes where multiple objects may produce overlapping or intertwined effects (*e.g.*, shadows and reflections), our method can consistently remove the objects but may sometimes fail to precisely eliminate the associated effects. This results in either (a) *under-removal of the target effect* or (b) *over-removal of effects belonging to nearby objects*.

## C.9 MORE COMPARISONS ON IN-THE-WILD DATA

In this subsection, we compare ObjectClear with state-of-the-art methods, including Power-Paint (Zhuang et al., 2024), Attentive Eraser (Sun et al., 2025), DesignEdit (Jia et al., 2025), RORem (Li et al., 2025), and OmniEraser (Wei et al., 2025), on in-the-wild data. To ensure a fair comparison, we consider two settings: (1) All methods use the same object mask; (2) While other methods are provided with finely annotated object-effect masks, both our ObjectClear and OmniEraser use only the object mask. The results under these two settings are shown in Fig. 18 and Fig. 19, respectively. Under the first setting, when given the object mask only, our ObjectClear effectively removes the shadow and reflection associated with the target object. Other methods fail to remove these effects or hallucinate undesirable content. OmniEraser can partially remove object and effects, but sometimes fails to remove them completely or over-removes unintended regions. Under the second setting, while some previous methods can remove shadows when given the effect region, they often alter or remove the original background content undesirably. OmniEraser, which uses only the object mask, can also remove some object effects but sometimes modifies the background or fails to fully remove the object and its effects. In contrast, our ObjectClear removes the object effect accurately with the object mask only while preserving the background with high fidelity.

## D USE OF LARGE LANGUAGE MODELS

The large language models (LLMs), *i.e.*, GPT-4o and Gemini 2.5 Pro, are solely used for polishing some paragraphs in this paper for clarity of expression and avoidance of minor grammar errors. They were not involved in the design of the methodology, execution of experiments, analysis of results, or any other aspect of the scientific contribution.

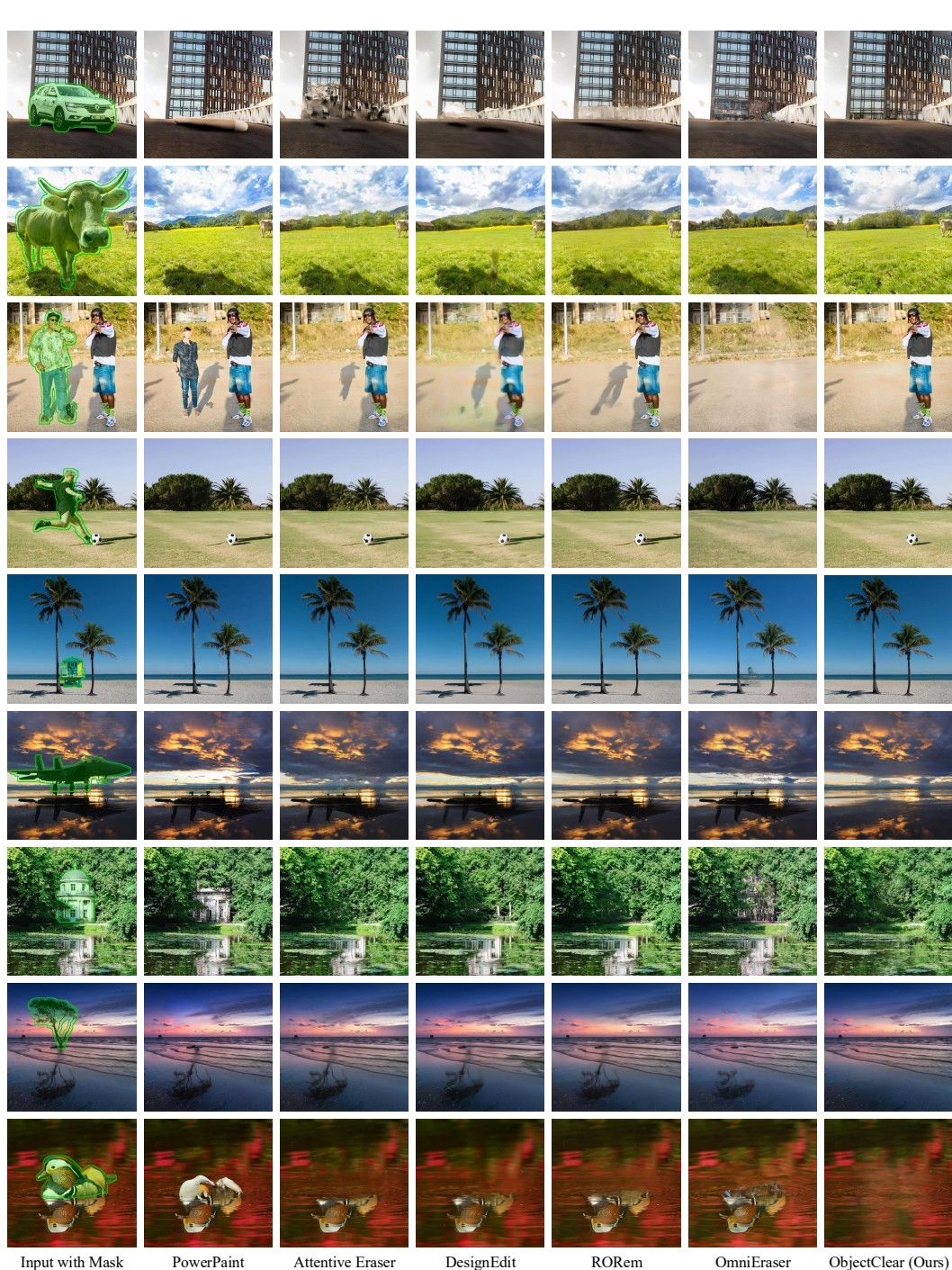

| Input with Mask | PowerPaint | Attentive Eraser | DesignEdit | RORem | OmniEraser | ObjectClear (Ours) |

Figure 18: **Object Removal on OBER-Wild - Condition on Object Mask.** Given the object mask only, our ObjectClear effectively removes shadow and reflection associated with the target object, while all the other methods fail to remove those effects and tend to hallucinate undesirable content or create artifacts.

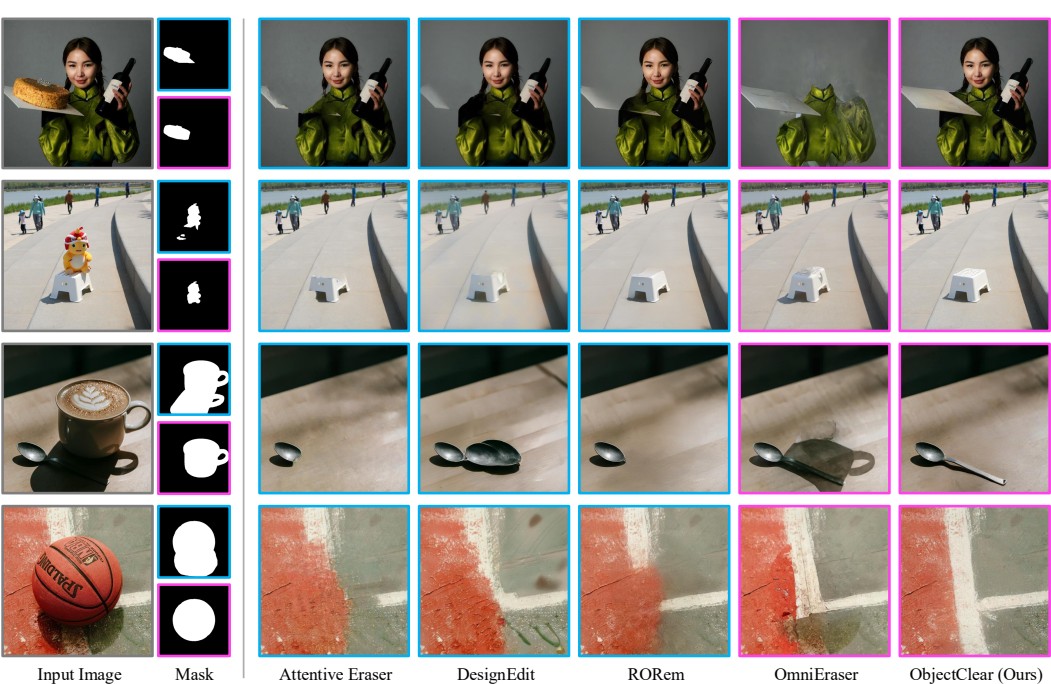

Figure 19: **Object Removal - Condition on** **Object-Effect Mask (others)** **and** **Object Mask (ours)**. Since some methods struggle to remove object effects when provided with only the object mask, we supply them with our annotated object-effect masks for a fair comparison. Although these methods are able to remove shadows with the additional effect region, they often introduce undesirable changes to the original background. In contrast, ObjectClear effectively removes the object and its associated effects using only the object mask, while preserving the background content with high fidelity.

