# OpenReview forum: "ObjectClear: Complete Object Removal via Object-Effect Attention"
_ICLR.cc/2026/Conference — ICLR 2026 Conference Withdrawn Submission_

### Official Review · Reviewer_fK3G · 2025-10-21

**Soundness:** 2
**Presentation:** 3
**Contribution:** 2
**Rating:** 2
**Confidence:** 5

**Summary:**

The paper introduces ObjectClear, a diffusion-based image editing framework tailored for object removal, with particular attention to preserving photorealism and secondary visual effects such as shadows and reflections. The proposed OBER dataset is a valuable contribution, offering both real and simulated image pairs, with and without object-induced effects, alongside precise object and effect masks. These comprehensive annotations support the design of an object-effect attention mechanism that aims to decouple foreground removal from background reconstruction. The results demonstrate improvements over existing baselines under evaluation. Moreover, the authors show that the proposed framework could be also used for object insertion and object movement.

**Strengths:**

- The paper introduces the OBER dataset, a new benchmark specifically designed for the object removal task. Each sample captures cases where removing an object also requires eliminating its associated effects such as shadows and reflections from the image, which addresses a more realistic and challenging scenario.
- The proposed dataset includes both real and simulated image pairs. For the simulated data, the authors develop a scalable generation pipeline capable of producing highly realistic images that incorporate object-induced effects, enhancing data diversity and controllability.
- The paper presents a novel diffusion-based editing model, termed ObjectClear, which integrates an object-effect attention mechanism to jointly handle object removal and the reconstruction of affected regions.
- In addition, the authors introduce OBER-Wild, a dataset of approximately 300 real-world images, to evaluate the model’s generalization performance under out-of-distribution conditions.

**Weaknesses:**

- The proposed model relies on an explicit object-effect attention mechanism, which necessitates training with image pairs containing precise masks for both objects and their associated effects. This requirement may limit scalability to datasets where such detailed annotations are unavailable.
- The evaluation primarily focuses on the RORD and OBER datasets. However, more recent benchmarks, such as OmniPaint-Bench (Yu et al., ICCV 2025), are not included, reducing the completeness of the empirical evaluation.
- In the out-of-distribution experiments, the authors employ the ReMOVE metric. As discussed by Yu et al. (ICCV 2025), this metric is not robust to object hallucinations. Incorporating more reliable metrics such as CFD, which aligns better with human perceptual judgments, would provide a stronger evaluation.
- The quantitative evaluation on RORD and OBER-Test omits several widely used metrics in image editing, such as FID, CMMD, and SSIM. Additionally, reporting no-reference metrics like ReMOVE and CFD for these experiments would help ensure a more comprehensive comparison.
- The Appendix demonstrates that the proposed model can also perform object insertion and movement, similar to the recently proposed OmniPaint model. However, no qualitative or quantitative comparisons are provided for these tasks, making it difficult to assess the model’s flexibility and its effectiveness in these additional applications.
- In Fig. 5 (sixth row), the shadow of the removed person remains visible on the wall in ObjectClear’s output. This example suggests potential limitations of the model when handling challenging cases involving occluded or complex object interactions.

**Questions:**

- The evaluation excludes the recently introduced OmniPaint-Bench dataset (Yu et al., ICCV 2025). Could the authors clarify whether there were any technical or compatibility constraints preventing its inclusion?
- Similarly, could the authors provide comparisons against the recent OmniPaint model to more clearly demonstrate ObjectClear’s relative effectiveness?
- Would the authors consider re-evaluating the model using additional metrics such as CFD, FID, CMMD, and SSIM? Including these results would provide a more comprehensive and fair comparison with prior work.
- If the model is indeed capable of object insertion and movement, these results would be valuable to include in the main paper, as they highlight the model’s broader editing capabilities. However, such results should be accompanied by more thorough qualitative and quantitative evaluations to support the claims.
- The paper presents only a few examples involving occlusions, where residual shadows or incomplete removals are observed. It would strengthen the paper if the authors could include more cases and analyses of such challenging scenarios in a revised version during the rebuttal phase.

**Details Of Ethics Concerns:**

None.

---

> ### Author Response · Authors · 2025-11-13
>
> We appreciate the reviewer’s efforts. However, several key weaknesses listed in the review are factually incorrect or based on misunderstandings, which we respectfully clarify below.
>
> **1. Comparison with OmniPaint was impossible at submission time**
>
> The review lists lack of comparison with OmniPaint (Yu et al., ICCV 2025) as a major weakness.
>
> However, the `OmniPaint inference code was publicly released only on Oct 19, 2025, which is nearly one month after the ICLR submission deadline (Sep 24, 2025)`:
>
> OmniPaint repo: https://github.com/yeates/OmniPaint/tree/main/scripts
>
> As the code was not available at submission time, this cannot be considered a valid weakness.
>
> **2. Evaluation on OmniPaint-Bench was also impossible at submission time**
>
> The review also lists the absence of experiments on OmniPaint-Bench (Yu et al., ICCV 2025) as a major weakness.
> Similarly, this `OmniPaint-Bench was released on Oct 19, 2025, again almost one month after the ICLR submission deadline (Sep 24, 2025)`.
>
> Dataset release:
> https://huggingface.co/datasets/yeates/omnipaint-bench/tree/main
>
> This benchmark did not exist at the submission time, so this weakness is also not valid.
>
> **Two key weaknesses are based on resources released after the submission deadline, these points should not be considered valid grounds for a low rating.**
>
> **3. Clarification about object–effect mask scalability**
>
> The review raises a concern that our object–effect masks “may limit scalability due to needing precise annotations.” We would like to clarify that this may stem from a misunderstanding of our data construction pipeline. As shown in Fig. 2(a) and Lines 185-186, once paired images (with and without the object) are available, the object–effect masks are produced via a simple thresholded difference, requiring no precise manual annotations and posing no scalability issue in practice.

---

### Official Review · Reviewer_JnQb · 2025-10-25

**Soundness:** 3
**Presentation:** 3
**Contribution:** 3
**Rating:** 6
**Confidence:** 4

**Summary:**

This paper addresses the incomplete nature of object removal, where existing methods often fail to eliminate associated visual effects like shadows and reflections (object-effects). The authors propose a comprehensive solution consisting of a new dataset, OBER, and a novel model, ObjectClear.

The OBER dataset provides high-quality paired images with and without object-effects, alongside precise masks for both the object and its effects, covering diverse scenarios.

The ObjectClear model utilizes a conditional diffusion inpainting framework incorporating two key innovations: (1) Dual-Mask Conditioning using two separate input masks for the object and its effect, providing explicit guidance; and (2) Object-Effect Attention (OEA), a proposed attention mechanism that explicitly models the interaction between object features and object-effect features, allowing the model to accurately reason about and remove the dependent effects. ObjectClear achieves state-of-the-art performance in complete object removal across various metrics and qualitative evaluations.

**Strengths:**

1. Crucial Dataset Contribution (OBER): The OBER dataset is a major strength. By explicitly providing object-effect ground truth, it enables specialized training and evaluation that previous inpainting datasets could not support.
2. Targeted Model Innovation (OEA): The Object-Effect Attention (OEA) module is highly effective. It is a smart architectural choice to address the core problem by establishing explicit cross-attention between object-related features and effect-related features.
3. Complete Solution: The paper offers a complete, end-to-end solution (data, task definition, and model) that demonstrably outperforms baselines on the specific, hard task of complete object removal, particularly excelling in handling complex shadows and reflections.

**Weaknesses:**

1. Ablation Complexity: While the ablation study is good, a clearer separation of the contributions from Dual-Mask Conditioning versus the OEA module would be valuable. Specifically, if the model used the dual masks but only standard self-attention, how much lower would the performance be compared to the full OEA setup?
2. The method focuses on shadows and reflections. A brief discussion on whether ObjectClear can implicitly handle other object-effects (e.g., subtle water ripples, dust cloud disturbances, or deformation effects) would clarify the module's general representational power.

**Questions:**

1. For better intuition, could the authors include a visualization (e.g., in the appendix) of the feature maps before and after the OEA module, potentially highlighting the regions corresponding to the object-effect features being refined or suppressed?
2. Are the shadows/reflections in OBER limited to simple, hard-edged examples, or do they include complex interactions like soft-edged shadows, multiple light sources, and distorted reflections on non-flat surfaces? Clarifying the diversity here would be helpful for understanding the model's generalization scope.

---

### Official Review · Reviewer_wSy1 · 2025-11-01

**Soundness:** 3
**Presentation:** 3
**Contribution:** 3
**Rating:** 6
**Confidence:** 3

**Summary:**

The paper proposes a new dataset and training/sampling strategy for object removal, focused on not only removing the identified (masked) object but also any remaining effects from the object such as shadows and reflections. The dataset consists of both camera-captioned real images and synthetically generated images by blending foreground objects with background images. At train time a cross-attention mask loss is introduced to train the network to focus on both object and object effects. At test time, the cross-attention mask is used to only update the image within the predicted mask. The evaluation shows that the approach outperforms other baselines.

**Strengths:**

The paper proposes a new dataset which consists of both real and synthetic images. The real images are valuable and labor intensive to create and the synthetic image pipeline seems to be well thought out and deliver good images.
The training approach with the mask loss also seems novel albeit a minor change, the same holds true for the sampling pipeline. The evaluation shows that the resulting model does indeed seems to effectively remove object effects which is something that existing models struggle with.

**Weaknesses:**

It would also be interesting to see an evaluation of the approach on a "general" object removal dataset to see if there are any negative effects when the objects do not show any shadows or reflections.
It would also be useful to do an evaluation comparison with the baselines without applying AG fusion, since that will by design improve the metrics by mostly taking original pixels. Running one or two of the better baselines with AG fusion (even though it might not remove the effect in that case) applied to them would be an interesting comparison, to see if it works at all.

**Questions:**

Did you play around with the attention maps and losses?
How robust is the method to the object mask? E.g., would it still work with a coarse object mask or does the approach need an accurate mask?
Will the dataset be open-sourced?

---

### Official Review · Reviewer_z5JN · 2025-11-01

**Soundness:** 2
**Presentation:** 3
**Contribution:** 2
**Rating:** 2
**Confidence:** 3

**Summary:**

The paper tackles complete object removal: deleting the target and its associated visual effects, such as shadows and reflections while keeping the background intact and without producing artifacts.  This paper presents new dataset  which provides paired images with and without object effects, along with precise masks for both objects and their effects. Then it proposes a method that incorporates an object-effect attention mechanism to guide the model toward the foreground removal regions by learning attention masks.

It shows performance gain over recent baselines, but reports failure in scenes with overlapping/entangled effects.

**Strengths:**

The proposed idea explicitly supervises attention on object-effect regions and reuses that attention at inference to protect the background, and shows clear performance empirically.

The paper is very clearly written and dataset construction is explained in detail.

**Weaknesses:**

This paper is centered  on dataset + slight change in architecture driven from practice.   Many recent works as mentioned in the related work section already tackle object-effect removal with datasets and tailored pipelines.  The core claimed contributions seems like a minor improvement over existing inpainting.
 Because it does not provide much learning insight or analysis, it might be better in computer vision/graphics conferences.

**Questions:**

Provide results and more discussions for OEA without mask supervision, AGF without OEA, SVDS vs uniform denoising, with statistics across datasets.

Sensitive to user masks ?

Public release ?

---

### Note · Authors · 2025-11-13

**Comment:**

We thank all reviewers for their efforts and thoughtful comments.

**Withdrawal Confirmation:**

I have read and agree with the venue's withdrawal policy on behalf of myself and my co-authors.